# Snail promotes ovarian cancer progression by recruiting myeloid-derived suppressor cells via CXCR2 ligand upregulation

Mana Taki[1], Kaoru Abiko[1], Tsukasa Baba[1], Junzo Hamanishi[1], Ken Yamaguchi[1], Ryusuke Murakami[1], Koji Yamanoi[1], Naoki Horikawa[1], Yuko Hosoe[1], Eijiro Nakamura[2], Aiko Sugiyama[2], Masaki Mandai[1], Ikuo Konishi[1] & Noriomi Matsumura[1]

Snail is a major transcriptional factor that induces epithelial-mesenchymal transition (EMT). In this study, we explore the effect of Snail on tumor immunity. Snail knockdown in mouse ovarian cancer cells suppresses tumor growth in immunocompetent mice, associated with an increase of $CD8^+$ tumor-infiltrating lymphocytes and a decrease of myeloid-derived suppressor cells (MDSCs). Snail knockdown reduces the expression of CXCR2 ligands (CXCL1 and CXCL2), chemokines that attract MDSCs to the tumor via CXCR2. Snail upregulates CXCR ligands through NF-kB pathway, and most likely, through direct binding to the promoters. A CXCR2 antagonist suppresses MDSC infiltration and delays tumor growth in Snail-expressing mouse tumors. Ovarian cancer patients show elevated serum CXCL1/2, which correlates with Snail expression, MDSC infiltration, and short overall survival. Thus, Snail induces cancer progression via upregulation of CXCR2 ligands and recruitment of MDSCs. Blocking CXCR2 represents an immunological therapeutic approach to inhibit progression of Snail-high tumors undergoing EMT.

[1] Department of Gynecology and Obstetrics, Kyoto University Graduate School of Medicine, 54 Shogoin Kawahara-cho, Sakyo-ku, Kyoto 606-8507, Japan. [2] DSK Project, Medical Innovation Center, Kyoto University Graduate School of Medicine, 53 Shogoin Kawahara-cho, Sakyo-ku, Kyoto 606-8507, Japan. Correspondence and requests for materials should be addressed to K.A. (email: kaoruvc@kuhp.kyoto-u.ac.jp)

Emerging evidence suggests that the acquisition of invasiveness in cancer is accompanied by the loss of epithelial features and the gain of a mesenchymal phenotype, a process known as epithelial-to-mesenchymal transition (EMT)[1,2]. In previous reports, gene expression clustering in ovarian cancer showed that the mesenchymal subtype, comprising enriched EMT-related gene signatures, had poor survival compared to other subtypes[1,3,4]. We discovered that there is decreased number of intraepithelial CD8+ tumor-infiltrating lymphocytes (CD8+TILs) in the mesenchymal subtype[4]. Thus, immune evasion might be taking place in the tumor undergoing EMT, although the mechanism of suppression of anti-tumor immunity in the state of EMT remains unclear.

Immune evasion is one of the major hallmarks of cancer[5], often accomplished via the recruitment of immunosuppressive cells such as myeloid-derived suppressor cells (MDSCs) or through the expansion of an immune checkpoint signal, namely the programmed death 1 (PD-1)/PD-1 ligand 1 (PD-L1) axis[6–11]. MDSCs represent a heterogeneous immature immunosuppressive myeloid cell population that expands during cancer progression and has the remarkable ability to suppress T cell functions in the tumor microenvironment[7]. Since MDSCs were officially described in 2007, an increasing number of studies have reported the biological and clinical significance of MDSCs[6,7,12,13]. We previously reported that MDSC infiltration was inversely correlated with CD8+TIL numbers and shorter overall survival in advanced ovarian cancer[14]. These reports indicated that MDSCs, as key players in cancer immune evasion, might be used both as prognostic factors and as therapeutic targets in cancer treatment.

Here, we focus on Snail, a key transcriptional repressor of E-cadherin during EMT[15,16], and explore the influence of Snail on MDSC infiltration into ovarian tumors. Elucidating the

mechanisms of Snail-induced immune evasion leads to the potential development of novel treatment strategies for tumor undergoing EMT.

## Results

**Snail is correlated with EMT and prognosis in ovarian cancer.** We first analyzed the dataset of high-grade serous ovarian cancer (HGSOC) from The Cancer Genome Atlas (TCGA) ($n = 266$). Application of a generic gene signature of EMT[3] to score the EMT status of the TCGA samples revealed that Snail expression was positively correlated with EMT scores (Fig. 1a). TCGA samples were divided into four molecular subtypes (differentiated, immunoreactive, mesenchymal, and proliferative) based on unsupervised hierarchical clustering using gene expression analysis[1]. Among the four subtypes, the mesenchymal subtype, which is associated with the poorest prognosis and has an EMT-related expression signature, had the highest Snail expression (Fig. 1b).

We next assessed the association between the EMT-related molecule Snail and prognosis in 56 cases of HGSOC (48 stage III cases and 8 stage IV cases) in Kyoto University Hospital. These samples were classified into four groups based on the nuclear expression of Snail, as determined by immunostaining (Fig. 1c). Samples with scores of 0/1 or 2/3 were assigned to the Snail-low and Snail-high groups, respectively. We ascertained the specificity of the anti-Snail antibody we used (Supplementary Figs. 1a, b). High Snail expression in the peritoneal dissemination was associated with shorter overall survival (Fig. 1d). Multivariate analysis using Cox proportional hazards regression indicated that Snail expression in the peritoneal dissemination serve as a negative, independent prognostic factor for overall survival (relative risk, 2.79; 95% confidence interval (CI), 1.10–7.05;

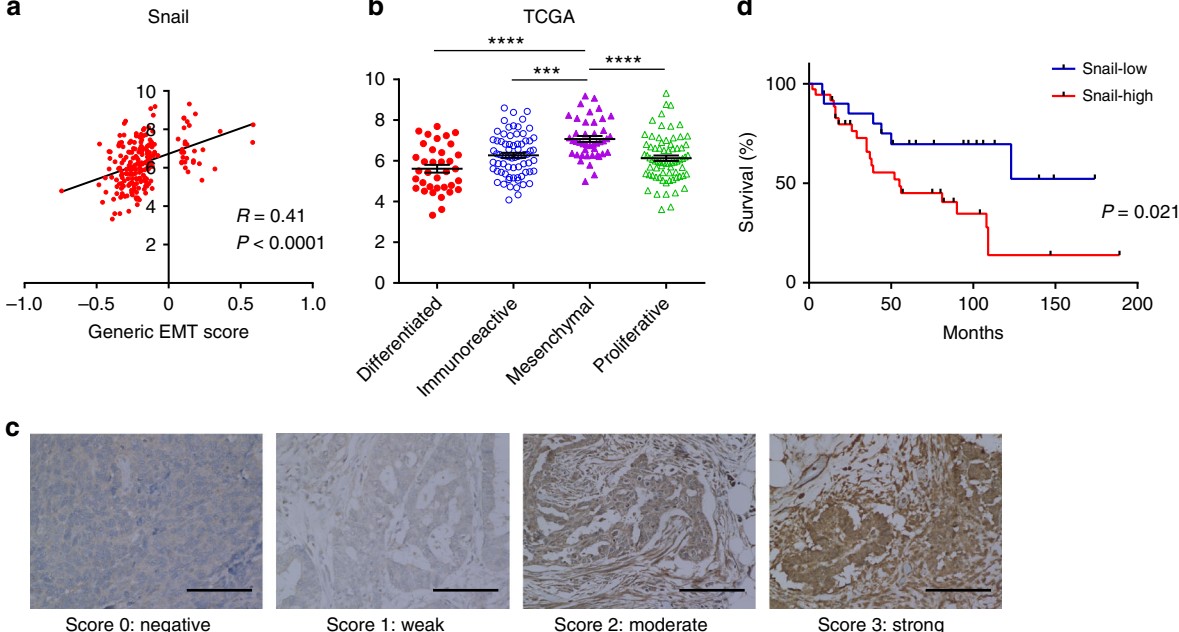

**Fig. 1** Snail expression is related to epithelial-to-mesenchymal transition and poor prognosis in ovarian cancer. **a** Correlation between generic epithelial-to-mesenchymal transition (EMT) scores for 266 high-grade serous ovarian cancer (HGSOC) samples from TCGA dataset[1] and Snail expression. The EMT signature was acquired from a previous report by Tan et al.[3]. Generic EMT scores were calculated following the method of single-sample Gene Set Enrichment Analysis (ssGSEA). The correlation coefficient ($R$) and $P$-value were based on Pearson's product-moment correlation analysis; $R = 0.41$ $P < 0.0001$. **b** Snail expression in HGSOC samples representing four subtypes from the TCGA datasets; ***$P < 0.001$ and ****$P < 0.0001$, mesenchymal vs. the other three subtypes based on one-way ANOVA with Tukey's multiple comparisons test. Bars: mean and SEM. **c** Classification of HGSOC as determined by the intensity of nuclear Snail immunostaining. Cases with scores of 0/1 were assigned to the Snail-low group, whereas those with scores of 2/3 were assigned to the Snail-high group to be used in survival analysis. Scale bars, 100 μm. **d** Overall survival curves for patients with HGSOC in Kyoto University Hospital ($n = 56$; 48 stage III cases and 8 stage IV cases), based on Snail staining of disseminated tumors in the omentum. $P = 0.021$ by log-rank test

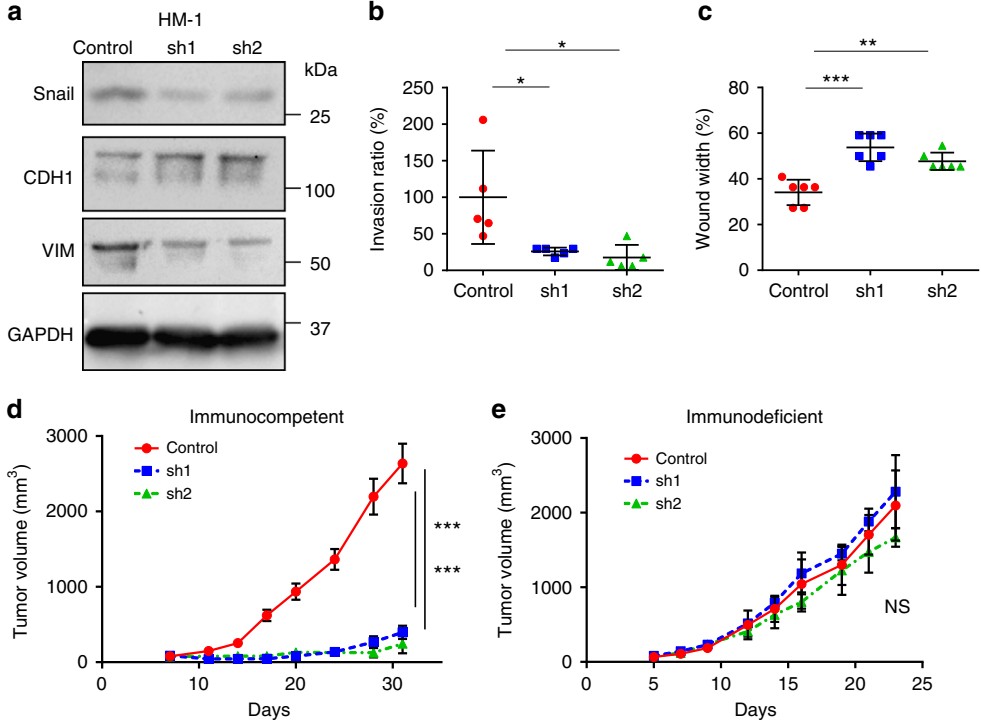

**Fig. 2** Snail suppresses tumor progression and is associated with tumor immunity. **a** Western blot of HM-1-control and HM-1-shSnail cells. CDH1: E-Cadherin, VIM vimentin. **b** Invasion of HM-1-control and HM-1-shSnail cells; $n = 5$. **c** Wound healing of HM-1-control and HM-1-shSnail cells. Data represent the percentage of wound width (24 h) relative to that at 0 h; $n = 6$. **d** Growth curves of HM-1-control and HM-1-shSnail subcutaneous tumors in immunocompetent mice. $P$-values represent significance between two groups at day 31; $n = 6$. **e** Growth curves of HM-1-control and HM-1-shSnail subcutaneous tumors from immunodeficient nude mice; $n = 6$; NS no significant difference between two groups at day 23. $*P < 0.05$, $**P < 0.01$, and $***P < 0.001$ (one-way ANOVA with Tukey's multiple comparisons test in **b**–**e**). Averaged data are presented as the mean ± SEM

$P = 0.031$; Supplementary Table 1). These results suggest that Snail expression in the peritoneal dissemination is associated with EMT and unfavorable prognosis in ovarian cancer.

**Snail induces EMT and promotes tumor progression in mice.** To elucidate the function of Snail, we generated HM-1-shSnail cell lines through the stable transduction of Snail-targeting two separate shRNA vectors into the HM-1 mouse ovarian cancer cell line. In HM-1-shsSnail cells, the expression of E-cadherin (an epithelial marker) increased, whereas that of vimentin (a mesenchymal marker) decreased (Fig. 2a). Cell invasion, assessed by wound-healing and invasion assays, was suppressed in HM-1-shSnail cells, compared to that in HM-1-control cells (Fig. 2b, c). Snail knockdown did not affect in vitro cell proliferation (Supplementary Fig. 2a). These data indicate that depletion of Snail inhibits EMT-like characteristics.

We also established Snail-depleted (OVCAR8-shSnail) and Snail-overexpressing (OVCA433-Snail) human ovarian cancer cell lines. Based on the altered expression of EMT-related proteins and changes in cell invasion capacity, it was demonstrated that EMT was inhibited in OVCAR8-shSnail cells and promoted in OVCA433-Snail cells (Supplementary Figs. 2b, c). Altered Snail expression did not affect in vitro cell proliferation (Supplementary Fig. 2d). Altered Snail expression did not affect cell apoptosis (Supplementary Fig. 2e). Collectively, these results confirmed that Snail expression induces EMT in ovarian cancer cell lines.

Next, we injected HM-1-shSnail cells subcutaneously or intraperitoneally into immunocompetent mice. Snail depletion inhibited tumor growth and prolonged overall survival compared to those in control animals (Fig. 2d and Supplementary Fig. 3a). In contrast, there was no significant difference in tumor growth

or in overall survival between HM-1-control and HM-1-shSnail groups in immunodeficient mice (Fig. 2e and Supplementary Fig. 3b). These data suggested that Snail facilitates tumor progression through mechanisms that are related to tumor immunity.

**Snail increases MDSCs and suppresses anti-tumor immunity.** Microscopically, HM-1-shSnail tumors were associated with increased CD8+TILs and decreased infiltration by Gr-1-positive or CD11b-positive MDSCs (Gr-1 and CD11b, a mouse MDSC marker) (Fig. 3a and Supplementary Figs. 4a, b). Flow cytometric analyses of immune cells in subcutaneous tumors revealed that the number of CD8+TILs and the percentage of IFNγ-positive CD8+TILs significantly increased in HM-1-shSnail samples ($P < 0.001$, unpaired $t$-test), whereas the number of MDSCs significantly decreased ($P < 0.01$, unpaired $t$-test) (Fig. 3b). Regarding MDSC subpopulations, granulocytic-MDSCs (G-MDSCs) were decreased in HM-1-shSnail tumors compared to levels in control tumors; however, there was no significant difference in terms of the numbers of monocytic-MDSCs (M-MDSCs) and tumor-associated macrophages (TAMs) (Supplementary Fig. 4c). CD49b+ cells (CD49b, a mouse natural killer (NK) cell marker) were increased in Snail-depleted tumors (Supplementary Fig. 5), although Snail interference did not suppress tumor growth of immunodeficient nude mice with the functioning NK cells (Fig. 2e).

We next examined the relationship between Snail and MDSCs in clinical ovarian cancer samples in Kyoto University Hospital. Using immunostaining, we evaluated the expression of CD33, a marker of human MDSCs, in primary ovarian tumors and in peritoneal disseminations (Fig. 3c). Snail expression exhibited a

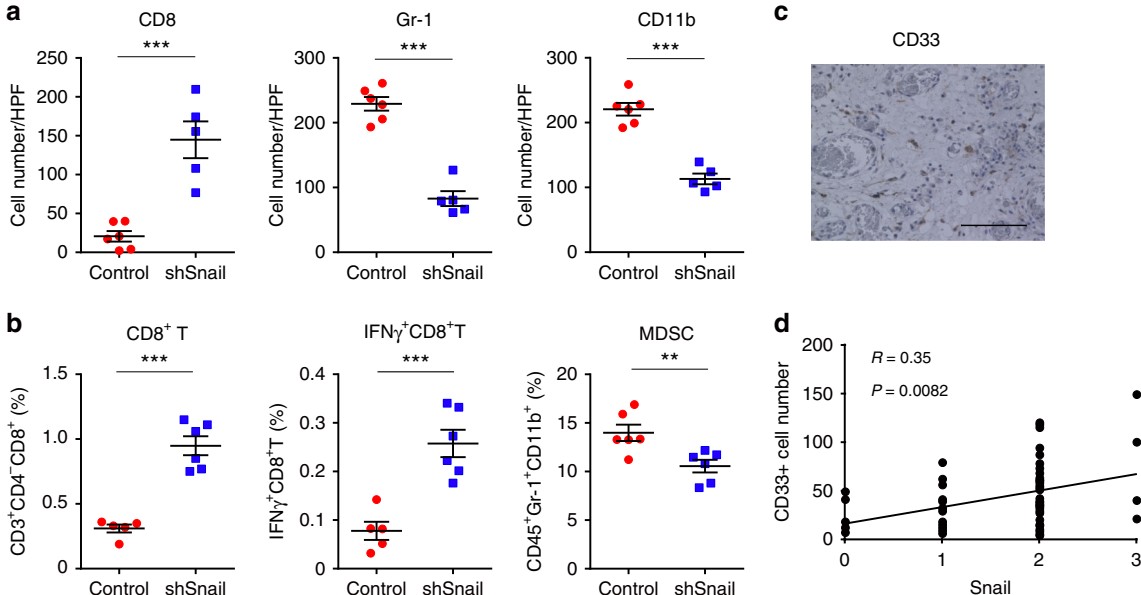

**Fig. 3** Snail is associated with increased intratumoral myeloid-derived suppressor cells. **a** Immunostained cell count in HM-1-control and HM-1-shSnail (sh1) subcutaneous tumors from immunocompetent mice. HPF high power field; $n = 5–6$. CD8$^+$ (left), Gr-1$^+$ (middle), and CD11b$^+$ (right). **b** Flow cytometry of HM-1-control and HM-1-shSnail (sh1) subcutaneous tumors from immunocompetent mice. The percentage of positive cells is plotted; $n = 6$. CD8$^+$ (left; CD3$^+$CD4$^−$CD8$^+$), intracellular IFNγ (middle; IFNγ CD3$^+$CD8$^+$), and MDSC (right; CD45$^+$Gr-1$^+$CD11b$^+$). **c** Representative image showing CD33 expression (a marker of human MDSC) in peritoneal dissemination of human high-grade serous ovarian cancer (HGSOC). Scale bar, 100 μm. **d** Correlation between infiltration of CD33$^+$ cells and Snail expression in corresponding disseminated tumors in the omentum ($R = 0.35$, $P = 0.0082$) of HGSOC; $n = 56$; Pearson's product-moment correlation analysis. $*P < 0.05$, $**P < 0.01$, and $***P < 0.001$ (unpaired $t$-test in **a** and **b**). Averaged data are presented as the mean ± SEM

significant correlation with the number of CD33$^+$ cells ($P = 0.0082$, Pearson's product-moment correlation analysis) (Fig. 3d). Human MDSCs are defined as CD33$^+$CD11b$^+$LIN$^−$HLA-DR$^{−/\text{low}}$ cells. Flow cytometry revealed that 85.1% of CD33$^+$ cells were confined to the CD33$^+$CD11b$^+$LIN$^−$ HLA-DR$^{−/\text{low}}$ type in ovarian tumor samples, and these cells expressed high levels of Arginase 1 regardless of HLA-DR expression (Supplementary Fig. 6). To examine whether intratumoral CD33$^+$ cells suppress T cell proliferation, we isolated CD33$^+$ cells from primary ovarian cancer samples and incubated the isolated cells with proliferating T cells. CD33$^+$ cells markedly inhibited T cell proliferation (Supplementary Fig. 7). Thus, CD33$^+$ cells were confirmed to represent MDSCs. These results demonstrated that Snail induces MDSC infiltration and suppresses anti-tumor immunity.

**CXCL1 and CXCL2 are expressed in Snail-high ovarian cancers.** To discover the factor responsible for increased intratumoral MDSCs in Snail-high tumors, we conducted microarray analysis of HM-1-control ($n = 3$) and HM-1-shSnail (sh1; $n = 2$ and sh2; $n = 2$) samples. This yielded 81 significantly downregulated genes and 19 significantly upregulated genes (changes of more than 2-fold and $P < 0.0005$ by one-way ANOVA; Supplementary Tables 2, 3). Particularly, CXCL2 and CXCL5 were downregulated more than 10-fold in HM-1-shSnail tumors (Fig. 4a). CXCL2 and CXCL5 are CXCR2 ligands. CXCR2 ligands have been reported to induce MDSC infiltration through CXCR2[17]. As expected, CXCR2 was highly expressed on MDSCs, especially G-MDSCs, in ascitic fluid samples from human ovarian cancer (Supplementary Figs. 8a, b).

Next, we assessed whether Snail alters CXCR2 ligand expression in human samples and cell lines. Based on TCGA data, Snail expression was correlated with the expression of CXCR2 ligands, specifically CXCL1, CXCL2 and, to a lesser extent, CXCL5 (Fig. 4b). Subsequently, we examined the

expression of CXCR2 ligands in human ovarian cell lines by reverse transcription polymerase chain reaction (RT-PCR). CXCL1, CXCL2, and CXCL5 expression was lower in OVCAR8-shSnail cells than in OVCAR8-control cells (Fig. 4c). OVCA433-Snail cells exhibited higher expression of CXCR2 ligands than OVCA433-control cells (Fig. 4c). Two more human ovarian cancer cell lines (A1847-shSnail and JHOS2-Snail) were used to validate that the chemokine levels were affected by Snail, and we obtained similar results (Supplementary Fig. 9). We also examined cytokine levels in supernatants of the human ovarian cancer cell lines by ELISA. Levels of CXCL1 and CXCL2 in OVCAR8-shSnail cells were lower than those in OVCAR8-control cells, whereas there was no significant difference between the two groups in terms of CXCL5 levels (Fig. 4d). CXCL1 and CXCL2 levels increased in OVCA433-Snail cells compared to those in OVCA433-control cells, whereas CXCL5 was not detectable in both groups (Fig. 4e).

Next, we assessed the expression of these chemokines in the mouse ovarian cancer cell line HM-1. RT-PCR showed that HM-1-shSnail cells expressed lower levels of Cxcl1, Cxcl2, and Cxcl5 (Fig. 4f). In subcutaneous mouse tumors, CXCL1 and CXCL2 levels were lower in HM-1-shSnail tumors, whereas CXCL5 levels were not different (Fig. 4g). CXCL1, CXCL2, and CXCL5 concentrations in the blood were lower in mice with HM-1-shSnail tumors (Fig. 4h). Together, these data demonstrate that Snail increases the expression of CXCL1 and CXCL2, chemokines known to induce MDSC infiltration.

**Snail induces CXCL1/CXCL2 expression via the NF-κB pathway.** To investigate the mechanism through which Snail induces CXCL1 and CXCL2 expression, we reevaluated the microarray data comparing HM-1-control to HM-1-shSnail tumors and analyzed the 81 downregulated genes by Gene Ontology (GO) enrichment analysis (Fig. 4a). The NF-κB pathway ($P < 8.88\text{e}^{-5}$)

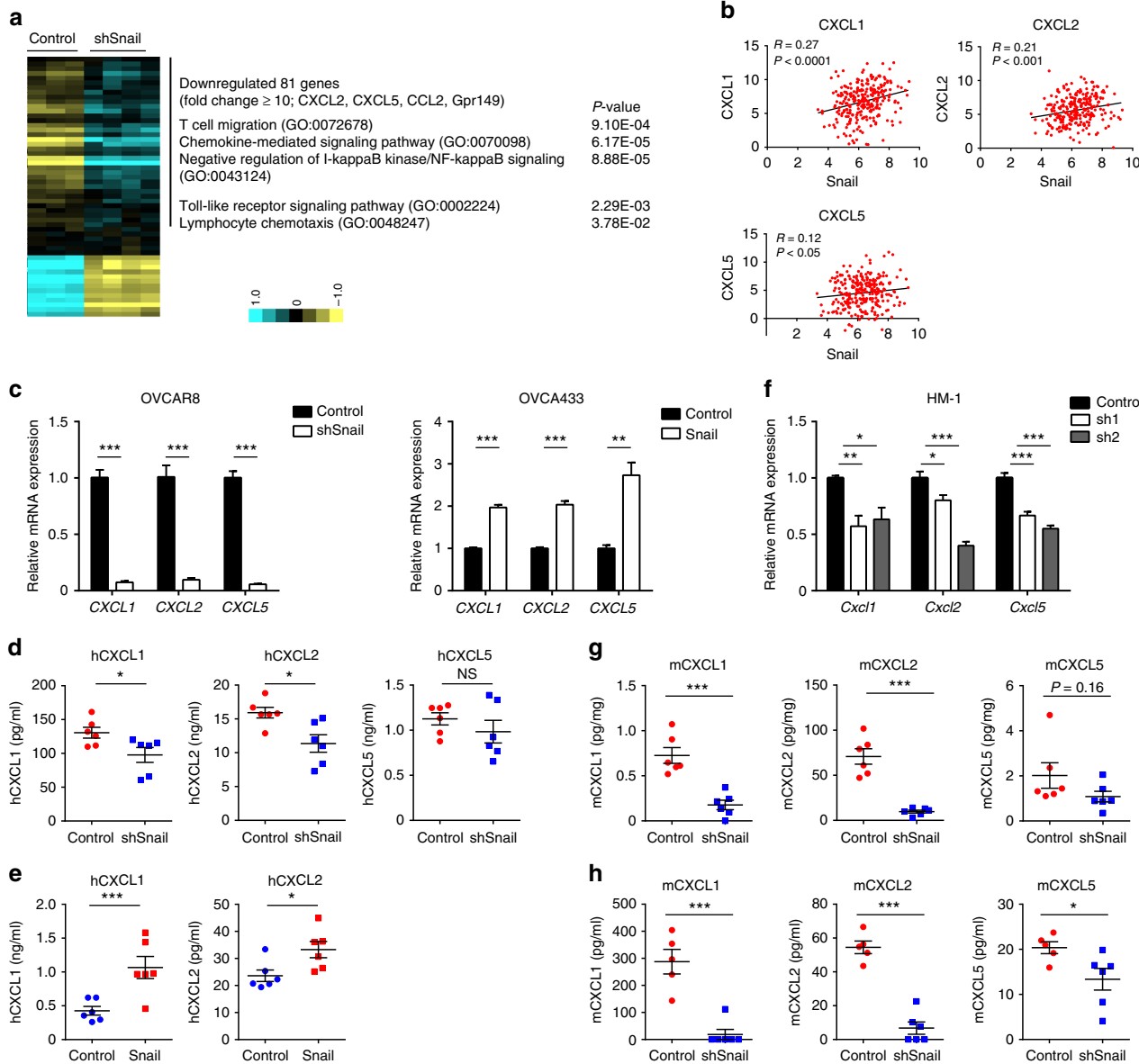

**Fig. 4** CXCR2 ligands are highly expressed in Snail-high ovarian tumors. **a** GO (Gene Ontology) term analysis of DNA microarray data obtained from HM-1-control (n = 3) and HM-1-shSnail (sh1; n = 2 and sh2; n = 2) cells. Genes that were significantly downregulated were used for GO term analysis. Genes are listed in Supplementary Table 2. **b** Snail expression was correlated with the expression of CXCR2 ligands (CXCL1; upper left, P < 0.0001, CXCL2; upper right, P < 0.001, and CXCL5; lower left, P < 0.05) in TCGA samples (n = 266). Pearson's product-moment correlation analysis. **c** Reverse transcription polymerase chain reaction (RT-PCR) of human ovarian cancer cell lines, OVCAR8 and OVCAR8-shSnail (left), and OVCA433 and OVCA433-Snail (right); n = 4. **d**, **e** ELISA of cell supernatants of human ovarian cancer cell lines, **d** OVCAR8 and OVCAR8-shSnail, and **e** OVCA433 and OVCA433-Snail; n = 6. **f** RT-PCR of HM-1-control and HM-1-shSnail cells; n = 5. **g**, **h** Levels of CXCL1, CXCL2, and CXCL5 in subcutaneous tumors (**g**) and in blood (**h**) taken from mice bearing HM-1-control and HM-1-shSnail tumors; n = 6. *P < 0.05, **P < 0.01, and ***P < 0.001 (unpaired t-test in **c**–**e**, **g**, **h**; one-way ANOVA with Tukey's multiple comparisons test in **f**). Averaged data are presented as the mean ± SEM

was among the downregulated pathways in HM-1-shSnail samples. It has been previously reported that the NF-κB pathway activates *CXCL1* and *CXCL2* gene transcription[18]. We hypothesized that the NF-κB pathway would be activated during Snail-induced EMT, which would enhance *CXCL1* and *CXCL2* expression. To further investigate this possibility, we analyzed nuclear phospho-p65 levels in ovarian cancer cell lines by western blot analysis. Phospho-p65, a main factor of the canonical NF-κB pathway, was decreased in Snail-depleted cell lines such as HM-1-shSnail and OVCAR8-shSnail, and was increased in the Snail-overexpressing cell line OVCA433-Snail (Fig. 5a). RelB, a factor involved in the non-canonical NF-κB pathway, was

unchanged in Snail-depleted cell lines. The results suggest that the canonical NF-κB pathway is activated by Snail. We next confirmed that NF-κB promotes *CXCL1* and *CXCL2* gene transcription. When treated with the NF-κB inhibitor, BAY11-7082, *Cxcl1* and *Cxcl2* expression decreased in HM-1-control cells (Fig. 5b). *CXCL1* and *CXCL2* expression also decreased in OVCAR8-control cells (Fig. 5b). However, a significant difference in *CXCL1* levels remained between BAY11-7082-treated control and Snail-depleted cells (P < 0.01, one-way ANOVA with Turkey's multiple comparisons test) (Fig. 5b), indicating that inhibition of the NF-κB pathway partially, but not totally, decreased *CXCL1* expression.

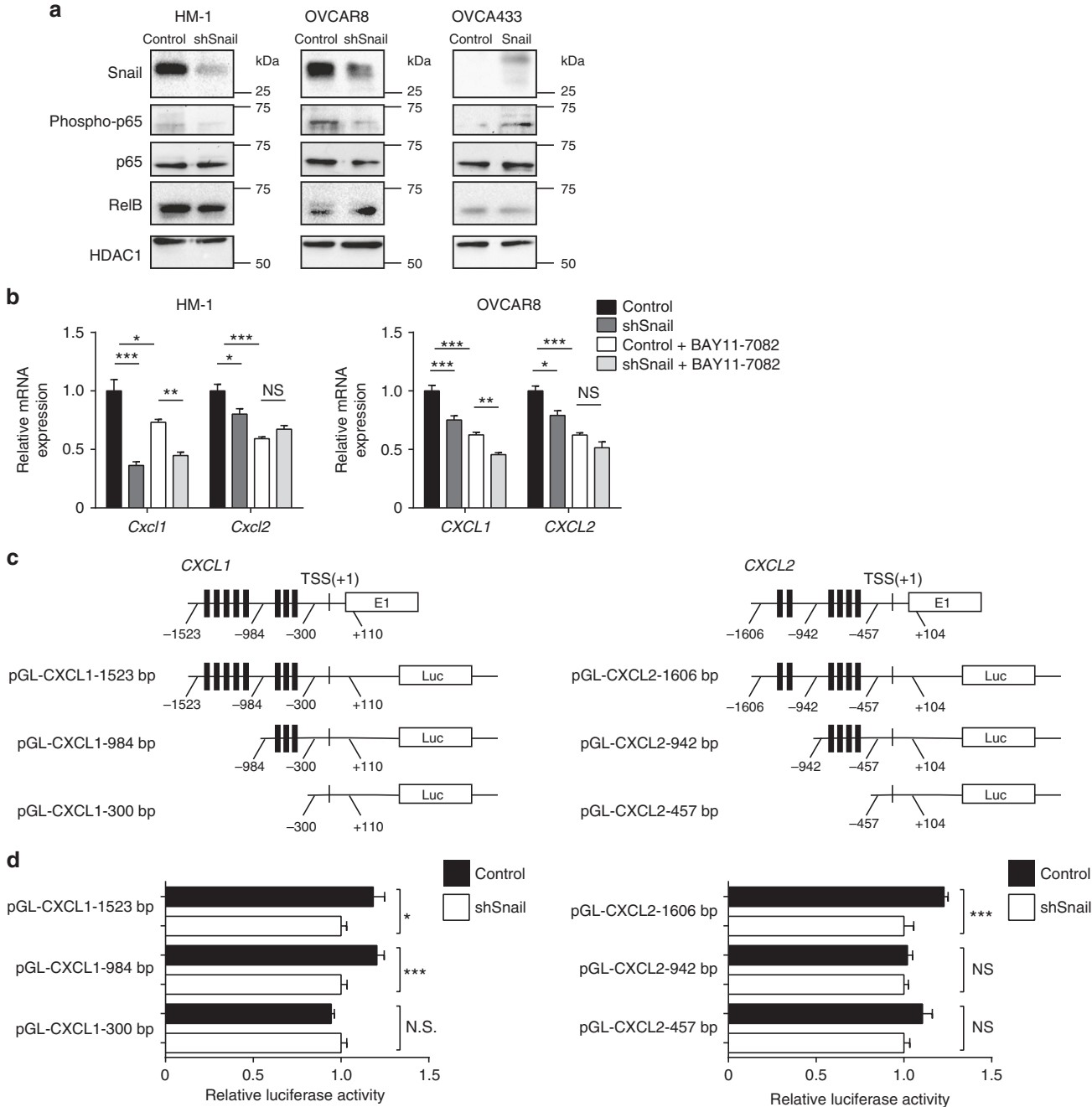

**Fig. 5** Snail induces CXCL1 and CXCL2 via the NF-κB pathway and possibly via the direct binding to their promoters. **a** Western blotting of nuclear Snail, phosphorylated p65 (phospho-p65), p65 and RelB from HM-1, OVCAR8 and OVCA433 cells. HDAC1 was used as control. **b** Reverse transcription polymerase chain reaction (RT-PCR) to analyze the expression of *Cxcl1* and *Cxcl2* in HM-1-control and HM-1-shSnail cells (left), and the expression of *CXCL1* and *CXCL2* in OVCAR8-control and OVCAR8-shSnail cells (right), treated with or without BAY11-7082 (NF-κB inhibitor) at 10 μM for 24 h; n = 4. **c** Schematic representation of *CXCL1* and *CXCL2* promoter organization, and the corresponding luciferase reporter constructs pGL-CXCL1 (1523 bp: −1523 to +110 bp, 984 bp: −984 to +110 bp, 300 bp: −300 to +110 bp) and pGL-CXCL2 (1606bp: −1606 to +104 bp, 942 bp: −942 to +104 bp, 457 bp: −457 to +104 bp). TSS transcriptional start site, E1 exon1, and Luc luciferase. The black bars indicate E-boxes (CANNTG), which are the binding sites of Snail. **d** Luciferase reporter assays to analyze the activity of the pGL-CXCL1 and pGL-CXCL2 promoter constructs in 293FT-control and 293FT-shSnail cells. Relative luciferase activities are shown; n = 5. *P < 0.05, **P < 0.01, and ***P < 0.001 (one-way ANOVA with Tukey's multiple comparisons test in **b**; unpaired *t*-test in **d**). Averaged data are presented as the mean ± SEM

We also considered the possibility of direct control of *CXCL1* and *CXCL2* gene transcription by Snail. A published dataset of chromatin immunoprecipitation sequencing (ChIP-seq) using an anti-Snail antibody (GSE61198) was analyzed to identify regions of increased sequence read tag density within the 5-kbp upstream and downstream sequences relative to the transcriptional start site

(TSS) of *Cxcl1* and *Cxcl2*[19]. It was revealed that Snail occupied the promoter of *Cxcl1*, and to a lesser extent *Cxcl2* (Supplementary Fig. 10). The binding of Snail appeared to activate *Cxcl1* and *Cxcl2* expression, because knockdown of Snail in HM-1 led to downregulation of *Cxcl1* and *Cxcl2*. To investigate this possibility using a luciferase reporter assay (as shown in Fig. 5c), we

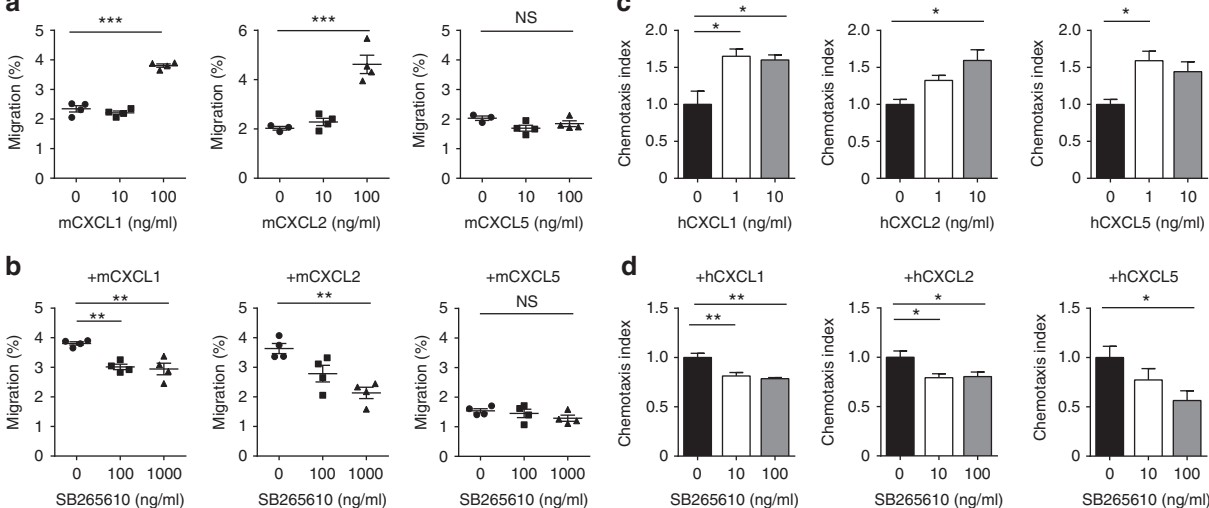

**Fig. 6** CXCR2 ligands induce myeloid-derived suppressor cell infiltration. **a** Chemotaxis of mouse MDSCs, from subcutaneous tumor of HM-1 tumor-bearing mice, in response to CXCL1, CXCL2, and CXCL5; $n = 4$. **b** Chemotaxis of mouse MDSCs, from subcutaneous tumor of HM-1 tumor-bearing mice, pre-treated with SB265610 (CXCR2 antagonist) at each concentration in the presence of each CXCR2 ligands (at 100 ng/mL); $n = 4$. **c** Chemotaxis response of MDSCs, from human ovarian cancer ascites, to CXCL1, CXCL2, and CXCL5. The chemotaxis index is shown; $n = 4$. **d** Chemotaxis response of human MDSCs treated with SB265610 at each concentration in the presence of each CXCR2 ligands; $n = 4$. *$P < 0.05$, **$P < 0.01$, and ***$P < 0.001$ (one-way ANOVA with Tukey's multiple comparisons test in **a–d**)

discovered E-boxes (CANNTG; binding sites for Snail) near their TSS and developed the corresponding luciferase reporter constructs (pGL-CXCL1-1523bp, 984 bp, 300 bp, pGL-CXCL2-1606bp, 942 bp, and 457 bp). Transfection of shSnail decreased the promoter activity of CXCL1-1523bp, CXCL1-984bp, and CXCL2-1606bp constructs (Fig. 5d), indicating that Snail might bind to the *CXCL1* promoter region from −984bp to −301bp and to the *CXCL2* promoter region from −1606bp to −943bp. These results suggest that Snail induces CXCL1 and CXCL2 expression through the NF-kB pathway, and possibly via direct transcriptional activation.

**CXCR2 ligands attract MDSCs and promote tumor growth.** Next, we examined the effect of CXCR2 ligands on MDSC chemotaxis. The migration of tumor-isolated MDSCs from HM-1-bearing mice was augmented by CXCL1 and CXCL2 in a dose-dependent manner (Fig. 6a). G-MDSCs were mainly attracted by CXCR2 ligands (Supplementary Fig. 11). Treatment with CXCR2 antagonist (SB265610) inhibited MDSC chemotaxis induced by chemokines (Fig. 6b). As in mouse MDSCs, MDSCs from human ovarian cancer ascitic fluid samples were attracted by CXCL1, CXCL2, and CXCL5, and a CXCR2 antagonist inhibited this migration (Fig. 6c, d). We made sure that MDSCs of tumor-bearing mice inhibited T cell proliferation (Supplementary Fig. 12). These data demonstrate that CXCR2 ligands induce MDSC chemotaxis through CXCR2.

In vivo G-MDSC depletion using an anti-Ly6G-antibody inhibited tumor growth in HM-1 tumor-bearing mice, but not in HM-1-shSnail tumor-bearing mice (Fig. 7a). MDSCs decreased and the CD8[+]T cell: MDSC ratio increased in anti-Ly6G antibody-treated HM-1-control tumors (Fig. 7b). G-MDSC depletion did not affect intratumoral MDSCs of HM-1-shSnail tumor-bearing mice (Fig. 7c). Moreover, in vivo treatment using a CXCR2 antagonist (SB265610) inhibited tumor growth in HM-1 tumor-bearing mice, but not in HM-1-shSnail tumor-bearing mice (Fig. 7d). MDSCs decreased and the CD8[+]T cell: MDSC ratio increased in SB265610-treated HM-1-control tumors (Fig. 7e). There were no significant differences in these parameters when HM-1-shSnail tumor-bearing mice were treated

with the CXCR2 antagonist (Fig. 7f). As expected, in a model of peritoneal dissemination, SB265610 significantly prolonged overall survival ($P = 0.016$, Log-rank test) (Supplementary Fig. 13). These results suggest that the CXCR2 antagonist reverse immunosuppression through the inhibition of G-MDSC migration to the tumor, thereby inhibiting tumor growth.

**CXCR2 ligand levels correlates with unfavorable prognosis.** Finally, we investigated the effect of CXCL1 and CXCL2 in clinical ovarian cancer samples. We measured the levels of CXCR2 ligands in serum samples from ovarian cancer patients ($n = 26$), and compared them to those of female healthy donors ($n = 8$). Ovarian cancer patients showed significant increases in both CXCL1 and CXCL2 compared to those in healthy donors (Fig. 8a). There was a significant correlation between CXCL1 and CXCL2 levels ($P = 0.0002$, Pearson's product-moment correlation analysis) (Fig. 8b).

We then sought to investigate the relationship between intratumoral MDSCs and serum CXCL1 and CXCL2 levels. Peritoneal disseminations ($n = 12$), from patients in which serum levels of CXCR2 ligands had been measured, were immunostained with CD33. Serum CXCL1 levels were associated with the number of CD33[+] cells (Fig. 8c). Serum CXCL2 levels were not associated with CD33 infiltration, perhaps due to the small sample number (Fig. 8c). There was no correlation between serum CXCR2 ligand levels and CD8[+], CD4[+], and FOXP3[+] cells (Supplementary Fig. 14). We further analyzed Snail expression by nuclear immunostaining in peritoneal dissemination samples. Snail expression showed a correlation with serum CXCL1 levels (Fig. 8d). MDSCs in PBMCs were also correlated with Snail expression in peritoneal dissemination samples and serum CXCL1 levels (Supplementary Fig. 15).

Finally, we investigated the association between serum concentrations of CXCR2 ligands and overall survival in ovarian cancer patients. Cases with high CXCR2 ligand levels were associated with a significantly poorer prognosis compared to those with low CXCR2 ligands (CXCL1, $P = 0.0005$, log-rank test, Fig. 8e; CXCL2, $P = 0.0008$, log-rank test, Fig. 8f). These results demonstrated that serum CXCL1 and CXCL2 levels predict

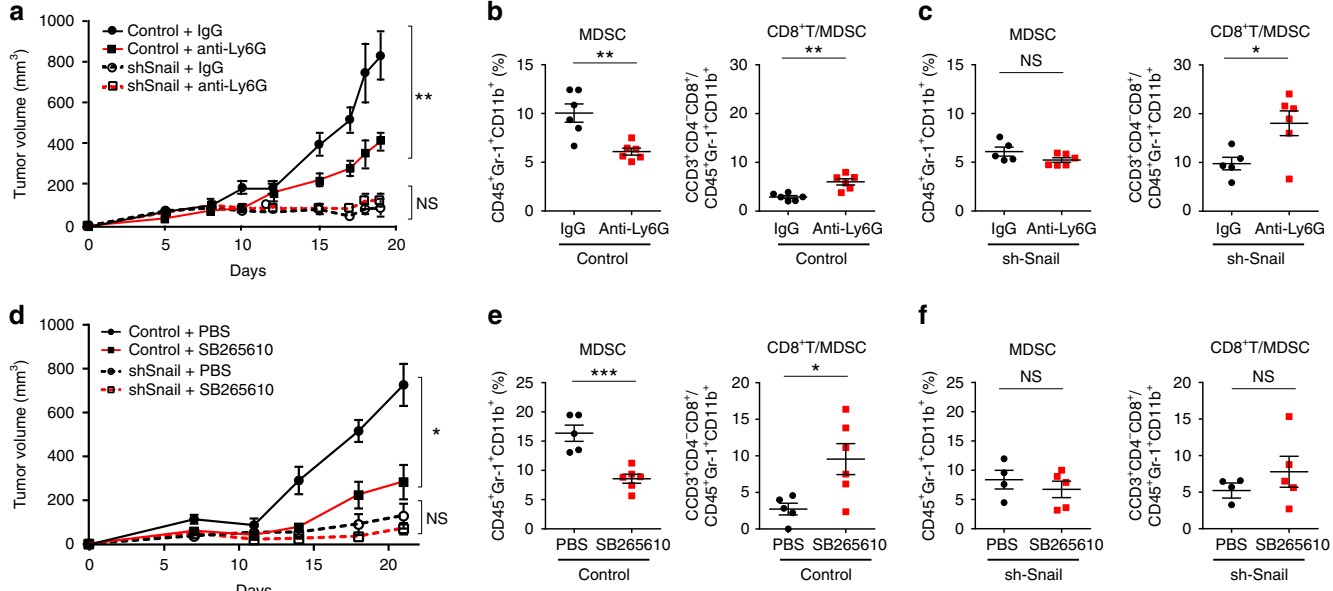

**Fig. 7** A CXCR2 antagonist inhibits tumor progression by Snail. **a** Tumor growth in mice subcutaneously injected with HM-1-control cells or HM-1-shSnail cells, treated with anti-Ly6G antibody (400 μg/body) or IgG twice a week from day 1 after tumor inoculation. *P*-values represent significance between two groups at day 19; *n* = 5–6. **b** Flow cytometric analyses of subcutaneous HM-1-control tumors treated with anti-Ly6G antibody or IgG at day 19. Percentage of positive cells relative to total cells is plotted. MDSCs (left) and CD8[+]T cells/MDSCs (right); *n* = 6. **c** Flow cytometric analyses of subcutaneous HM-1-shSnail tumors treated with anti-Ly6G antibody or IgG at day 19. The percentage of positive cells relative to total cell count is plotted. MDSCs (left) and CD8[+]T cells/MDSCs (right); *n* = 5–6. **d** Tumor growth in mice subcutaneously injected with HM-1-control cells or HM-1-shSnail cells, treated with SB265610 (2 mg/kg body weight) or PBS six times a week from day 1 after tumor inoculation. *P*-values represent significance between two groups at day 21; *n* = 4–6. **e** Flow cytometric analyses of subcutaneous HM-1-control tumors treated with SB265610 or PBS at day 21. Percentage of positive cells relative to total cells is plotted. MDSCs (left) and CD8[+]T cells/MDSCs (right); *n* = 5–6. **f** Flow cytometric analyses of subcutaneous HM-1-shSnail tumors treated with SB265610 or PBS at day 21. The percentage of positive cells relative to total cell count is plotted. MDSCs (left) and CD8[+]T cells/MDSCs (right); *n* = 4–5. *$P < 0.05$, **$P < 0.01$ and, ***$P < 0.001$ (unpaired *t*-test in **a**–**f**)

intratumoral MDSC infiltration and Snail expression, and are associated with poor prognosis.

## Discussion

EMT has been widely accepted as a key process for tumor invasion and metastasis, since the loose cell–cell contacts and high motility derived from EMT facilitate tumor cell detachment from basement membrane and tumor invasion into the surrounding stroma or vessels[16,20]. EMT affects various characteristics of the tumor such as stemness, cell–cell junction, apoptosis, neuronal differentiation, or immune system process[21]. Therefore, multiple chemical drugs targeting EMT have been developed for cancer therapy[22]. However, the relationship between EMT and immune evasion remained unclear.

Snail, a key transcriptional repressor of E-cadherin during EMT, has a pivotal role in cancer progression[15,16]. In the present study, the immunological effect of Snail on tumor microenvironment in ovarian cancer is explored. We showed that Snail induces intratumoral trafficking of MDSCs via upregulation of CXCR2 ligands. Surprisingly, ovarian cancer mouse model demonstrated that Snail facilitates tumor progression only in immunocompetent mice, indicating that the tumor-promoting mechanisms of Snail are largely related to tumor immunity (Fig. 2). In our microarray data, immune-associated pathways were downregulated by Snail knockdown (Fig. 4a). Analysis of TCGA ovarian cancer datasets indicates that Snail is correlated with many cytokines including CXCR2 ligands (Fig. 4b). Some reports have shown that Snail induces other immunosuppressive cells such as regulatory T cells or TAMs[23,24]. Our findings and these reports support the idea that Snail has role in multiple functions including immune system, and Snail is a crossroad of the two hallmarks of cancer, EMT, and immune evasion.

Here, we focused on CXCL1 and CXCL2, which function as chemokines to attract MDSCs to the tumor, since Snail knockdown attenuated intratumoral MDSC which suppressed CD8[+]T cell proliferation (Fig. 3, Supplementary Figs. 4, 12). Interestingly, MDSCs are also known to promote metastasis by inducing EMT[25]. Together with our data, intratumoral MDSC infiltration induced by Snail accelerates EMT and leads to further tumor progression. Blocking this malicious cycle, for example with a CXCR2 antagonist, would be a promising treatment strategy. We demonstrated that the CXCR2 antagonist SB265610 suppresses the recruitment of G-MDSCs to the tumor and thereby inhibits tumor progression (Fig. 7 and Supplementary Fig. 13). Since G-MDSCs mainly express CXCR2 in human samples (Supplementary Fig. 8), a CXCR2 antagonist could be an effective treatment for ovarian cancer patients. However, blockade of CXCR2 largely but not completely inhibited tumor growth, suggesting that Snail might activate other factors involved in antitumor immunity, and the issue requires further investigation.

Serum CXCL1 and CXCL2 levels are elevated in ovarian cancer patients and are associated with intratumoral MDSC infiltration (Fig. 8). Furthermore, serum CXCL1 and CXCL2 protein levels are associated with poor prognosis (Fig. 8). These data indicate that serum chemokines represent a measurable marker that reflects MDSC infiltration, and could be used to estimate immune status of the tumor microenvironment, in place of tumor specimens acquired by operation or biopsy. Some researchers have reported that the expression of CXCL1 and CXCL2 is correlated with tumor size, metastasis, and decreased overall survival in breast, gastric, colorectal, hepatocellular, and bladder carcinoma

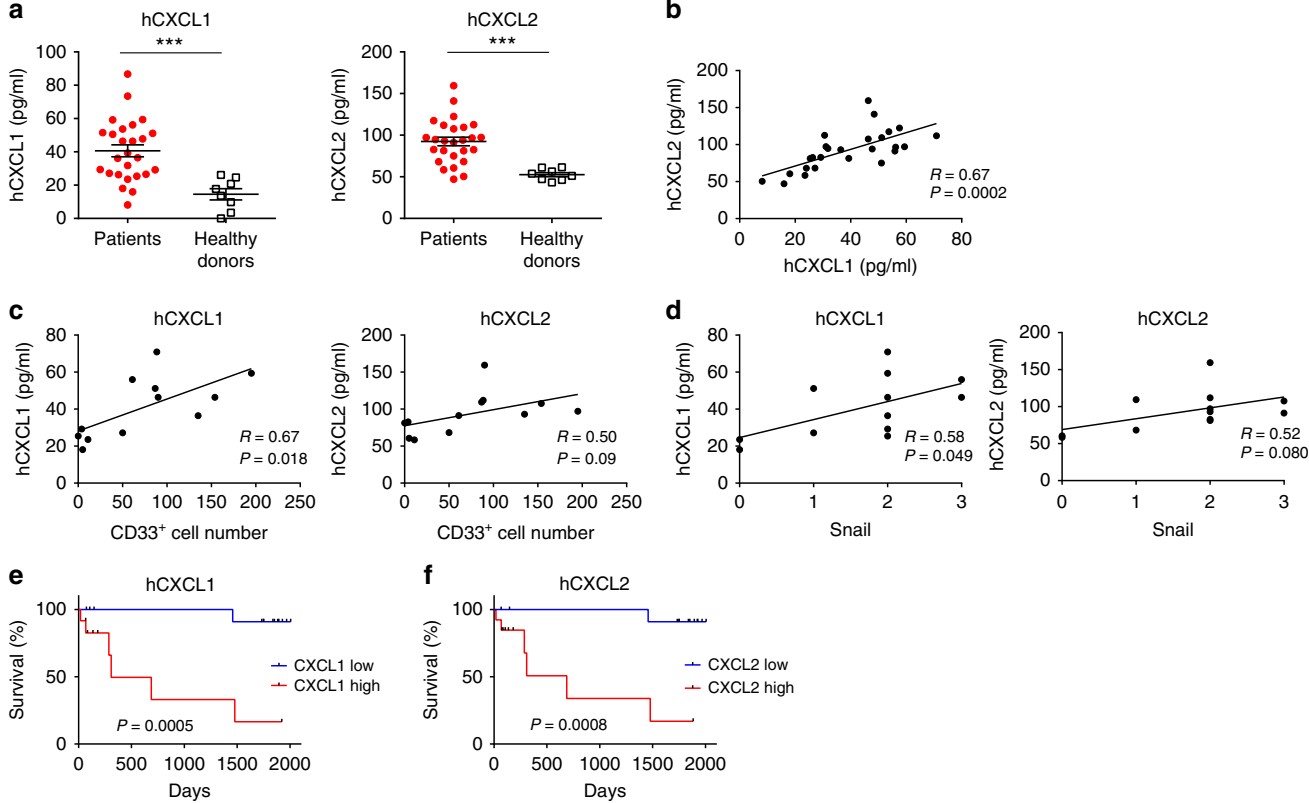

**Fig. 8** Elevated levels of serum CXCL1 and CXCL2 reflect intratumoral MDSCs and are associated with poor prognosis in ovarian cancer patients. **a** Serum concentration of CXCL1 (left) and CXCL2 (right) in ovarian cancer patients ($n = 26$) vs. that in healthy donors ($n = 8$); mean ± SEM; ***$P < 0.001$ by unpaired $t$-test. **b** Correlation between CXCL1 and CXCL2 serum concentrations in ovarian cancer patients ($n = 26$); $P = 0.0002$, $R = 0.67$. Pearson's product-moment correlation analysis. **c** Correlation between serum CXCL1 (left; $P = 0.018$) and CXCL2 (right; $P = 0.09$) levels and infiltration of CD33[+] cells in peritoneal disseminations; $n = 12$; Pearson's product-moment correlation analysis. **d** Correlation between serum CXCL1 (left; $P = 0.049$) and CXCL2 (right; $P = 0.080$) levels and Snail staining scores; $n = 12$; Pearson's product-moment correlation analysis. **e** Overall survival of ovarian cancer patients, comparing the high-serum CXCL1 group ($n = 12$) to the low-serum CXCL1 group ($n = 14$). Cut-off levels, 42.83 pg/ml; AUC = 0.6842. Hazard ratio, 15.08; 95% confidence interval (CI), 3.859–102.9; $P = 0.0005$ by log-rank test. **f** Overall survival of ovarian cancer patients, comparing the high-serum CXCL2 group ($n = 13$) to the low-serum CXCL2 group ($n = 13$). Cut-off levels, 93.57 pg/ml; AUC = 0.6767. Hazard ratio, 13.87; 95% confidence interval (CI), 3.56–89.72; $P = 0.0008$ by log-rank test

patients[26–30]. Although multivariate analysis failed to demonstrate that the expression of these chemokines is an independent predictor of unfavorable prognosis in ovarian cancer, elevated chemokine levels were related to Snail and MDSC infiltration (Fig. 8). To overcome the possible limitation of small sample size, it is necessary to conduct larger clinical studies in the future to evaluate serum CXCL1 and CXCL2 as biomarkers.

In conclusion, Snail upregulates the expression of CXCL1/2 and enhance MDSC tumor infiltration via CXCR2; MDSCs were shown to inhibit anti-tumor immunity and promote tumor progression. CXCR2 and its ligands could thus be a therapeutic target for tumors with high Snail expression. These findings provide new insights into the mechanism that might link EMT to tumor immunity.

## Methods

**Cell lines.** The OV2944-HM-1 (HM-1) mouse ovarian cancer cell line and JHOS2 were purchased from RIKEN BioResource Center and were cultured as previously described in ref. [14,31]. Human ovarian cancer cell lines (OVCAR8, OVCA433, and A1847) were provided by Dr. Susan K. Murphy from Duke University. These cells were cultured and maintained in RPMI1640 (Invitrogen, Carlsbad, CA, USA) supplemented with 10% fetal bovine serum, 100 U/mL penicillin, and 100 g/mL streptomycin in an atmosphere containing 5% $CO_2$ at 37 °C. 293FT cells were purchased from Thermo Fisher Scientific (Waltham, MA, USA) and cultured as described in ref. [32]. Throughout this study, we used cell lines that were passaged

fewer than 30 times. All cell lines were regularly tested for mycoplasma contamination.

**shRNA vectors and ORF.** HM-1-shSnail cells were generated by lentiviral transfection with short hairpin RNAs (shRNAs) targeting Snail using Mouse GIPZ Lentiviral shRNA viral particles (GE Healthcare UK, Buckinghamshire, England; sh1, clone ID V3LMM_523301, gene target sequence AGTTTATTGATATTCA ATA; sh2, clone ID V3LMM_523299, gene target sequence TGGTTAATTTA TATACTAA).

OVCAR8-shSnail, A1847-shSnail, and 293FT-shSnail cell lines were generated by Human GIPZ Lentiviral shRNA viral particles (GE Healthcare UK; clone ID V3LHS_328732, gene target sequence AGCGAGCTGCAGGACTCTA). Control cell lines, HM-1-control, OVCAR8-control, and 293FT-control, were generated through the transfection of a non-silencing, control shRNA (GE Healthcare UK; clone ID RHS4348).

The Snail-overexpressing cell lines, OVCA433-Snail, and JHOS2-Snail were generated by lentiviral transfection of an open reading frame using Precision LentiORF viral particles (GE Healthcare UK; clone ID PLOHS_100004229). The control cell lines, OVCA433-control, and JHOS2-control, were generated by transfecting a non-silencing, control shRNA (GE Healthcare UK; clone ID OHS5833).

**RNA extraction and real-time quantitative PCR.** Total RNA was extracted from cells and frozen tissues using an RNeasy Mini Kit (Qiagen, Venlo, Netherlands). A Transcriptor High Fidelity cDNA Synthesis Kit (Roche Diagnostics, Roswell, GA, USA) was used for cDNA synthesis. For RT-PCR, cDNA was amplified using a thermal cycler. RT-PCR was performed by amplification of the target genes and *Gapdh* mRNA as a reference gene using a Light Cycler 480-II (Roche Diagnostics). The primers used in these experiments are listed in Supplementary Table 4.

Relative target gene expression was estimated by dividing the threshold cycle (CT) value of the target gene by the CT value for *Gapdh* mRNA.

**Immunoblotting**. Cell pellets were lysed with 1× RIPA buffer (Thermo Fisher Scientific) containing a protease inhibitor cocktail (Nacalai Tesque, Kyoto, Japan) and a phosphatase inhibitor cocktail (Nacalai Tesque). Nuclear proteins were collected using NE-PER Nuclear and cytoplasmic extraction reagents (Thermo Fisher Scientific) according to the manufacturer's protocol. Proteins were subsequently separated by SDS-PAGE and transferred to nitrocellulose membranes. Membranes were immunoblotted with the following antibodies: Rabbit anti-NF-κB p65 (phospho S536) antibody (ab28856; 1:500 dilution; Abcam), rabbit anti-NF-κB p65 antibody (#8242; 1:1000 dilution; cell signaling), rabbit anti-RelB antibody (#4922; 1:1000 dilution; cell signaling), rabbit anti-SNAIL antibody (ab180714; 1:1000 dilution; Abcam), rabbit anti-E-cadherin antibody (sc-7870; 1:500 dilution; Santa Cruz), mouse anti-Vimentin antibody (MA5-11883; 1:200 dilution; Invitrogen), mouse anti-GAPDH antibody (ab9484; 1:1000 dilution; Abcam), and rabbit anti-HDAC1 antibody (ab109411 1:1000 dilution; Abcam). The bands were visualized using Molecular Imager Gel DocTMXR+ and ChemiDocTMXRS+ Systems with Image Lab 2.0 software (Bio-Rad). Uncropped scans of the blotted gels with the weight markers are shown in Supplementary Fig. 16.

**ELISA**. Tumor tissue was weighed, treated in a bead homogenizer, and sonicated in 1× RIPA buffer (Thermo Fisher Scientific) containing a protease inhibitor cocktail (Nacalai Tesque) and a phosphatase inhibitor cocktail (Nacalai Tesque), and was centrifuged at $16,000 \times g$ for 10 min at 4 °C. Mouse blood was obtained from tumor-bearing mice, and was allowed to clot for 30 min on ice, before being centrifuged at $16,000 \times g$ for 10 min at 4 °C and serum was aspirated. Samples were subjected to ELISA analysis. Mouse CXCL1, CXCL2, and CXCL5 protein levels were measured by mouse Quantikine ELISA kits for CXCL1, CXCL2, or CXCL5 (R&D systems, Minneapolis, MN, USA) according to the manufacturer's protocols. Human CXCL1, CXCL2, and CXCL5 protein levels in culture supernatants and patient serum samples were measured using Human ABTS ELISA Development Kits for CXCL1 or CXCL2 (PeproTech, Rocky Hill, NJ, USA) and a human CXCL5 Quantikine ELISA kit according to the manufacturer's protocols.

**Cell proliferation assays**. Water soluble tetrazolium-8 assays were performed using Cell Count Reagent SF (Nacalai Tesque, Kyoto, Japan) according to the manufacturer's protocol, and the proliferation rate for each cell line was calculated and plotted.

**Wound-healing assays**. Cells were seeded on 6-well plates and grown to confluency. Subsequently, the monolayer was gently scratched with a pipette tip to create a mechanical wound. Images were taken at 0 and 24 h using a microscope.

**Invasion assays**. Cells were seeded in Boyden chambers with 8.0-μm pore PET membranes (Becton Dickinson, Franklin Lakes, NJ, USA) that were coated with Matrigel. After 24 h, Boyden chamber membranes were fixed with 4% PFA, permeabilized with 99% methanol, and stained with hematoxylin. The number of cells that invaded through the membrane was visually counted at a magnification of ×200 from five fields and the average number was calculated as previously described in ref. [33]. The invasion rate for each cell line was calculated and plotted.

**Apoptosis assay**. Cells were seeded on 24-well plates and grown to confluency. Cells were harvested and analyzed by a MACS Quant (Miltenyi Biotec, Bergisch Galdbach, Germany) following staining with APC-labeled annexin V antibody (BD Pharmingen/BD Biosciences, Franklin Lakes, NJ, USA) and propidium iodide according to the manufacturer's protocol.

**NF-κB inhibitor treatment assays**. A total of $1 \times 10^5$ HM-1 cells in 24-well dishes or $1 \times 10^5$ OVCAR8 cells in 12-well dishes were treated with BAY11-7082 (Santa Cruz Biotechnology, Dallas, TX, USA) at 10 μM for 24 h. Total RNA was extracted from cells and frozen tissues using an RNeasy Mini Kit (Qiagen). RT-PCR was then performed as previously described.

**Luciferase reporter assays**. A DNA fragment for the human cytomegalovirus minimal promoter was inserted between the Bgl II and Hind III sites of pGL3 promoter vector (Promega, Madison, WI, USA), and the resultant pGL3/CMVmp plasmid was provided by Prof. Harada and Dr. Katagiri. The regulatory regions of the CXCL1/CXCL2 gene (−1523/−984/−300 to +110, or −1606/−942/−450 to +104, respectively, of the surrounding the TSS) were cloned by PCR amplification of genomic DNA and then inserted between the Kpn I and Xho I sites of the pGL3/CMVmp plasmid to generate CXCL1/CXCL2-Luc reporter constructs.

Luciferase reporter assays were performed by transfecting the reporter construct into 293FT-shSnail cells. The pRL-CMV vector was co-transfected in each experiment as an internal control for transfection efficiency. At 48 h post-transfection, luciferase activities were measured using the Dual-Luciferase Reporter Assay System (Promega) according to the manufacturer's protocols.

**Mice**. Female B6C3F1 (C57BL6×C3/He F1) mice were purchased from CLEA Japan (Tokyo, Japan) and used in immunocompetent mice experiments. Female ICR-nu (Crlj: CD1-*Foxn1^{nu}*) mice was purchased from Charles River Japan (Yokohama, Japan) and used in immunodeficient mice experiments. Animals were maintained under specific pathogen-free conditions.

**Animal experiments**. A total of $1 \times 10^6$ HM-1-shSnail (sh1) or HM-1-control cells were inoculated subcutaneously into the right flank or injected into the abdominal cavity of 6–8-week-old female B6C3F1 mice and nude mice. Anti-Ly6G antibody (clone 1A8; BioXcell, West Lebanon, NH, USA) or mouse IgG2a (clone 2A3; BioXcell) treatment was initiated 1 day after tumor cell inoculation and was administered intraperitoneally twice a week (400 μg per body) as previously describedin ref. [34]. CXCR2 antagonist treatment was initiated 1 day after tumor cell inoculation and was administered intraperitoneally six times a week (SB265610: 2 mg/kg body weight; R&D systems) as previously described in ref. [35]. Mice were euthanized before becoming moribund. The subcutaneous tumor size and body weight were measured twice a week and tumor volumes were calculated as follows: volume $= LD \times SD^2 \times 0.5$, where LD is the long diameter (mm) and SD (mm) is the short diameter of the tumor. Cages of mice were allocated to experimental groups by random draw. The investigator was not blinded to the group allocation. Sample sizes were chosen to assure reproducibility of the experiments in accordance with the replacement, reduction, and refinement principles of animal ethics regulation. All animal studies were approved by Kyoto University Animal Research Committee.

**Immunohistochemical analysis of mouse tumors**. Tumor cryosections (6-μm-thick) were stained with anti-CD8 (clone YTS169.4; Abcam), anti-Gr-1 (clone RB6-8C5; BD Pharmingen), or anti-CD11b (clone M1/70; BioLegend, San Diego, CA, USA) antibodies as previously described in ref. [14]. CD8+, Gr-1+, and CD11b+ cell infiltration into omental tumors was quantified and evaluated.

**Immunohistochemical analysis of human clinical samples**. Immunostaining was performed using the streptavidin-biotin-peroxidase method as previously described in ref. [9]. Double immunostaining was performed using MACH2 double stain polymer detection reagent (BRR525; BioCare Medical, CA, USA) according to the manufacture's protocol. For Snail, CD33, p53, CD8, CD4, and FOXP3, the samples were incubated with a rabbit anti-SNAIL polyclonal antibody (ab180714; 1:1000 dilution; Abcam, Cambridge, UK), mouse anti-CD33 monoclonal antibody (clone PWS44; 1:200 dilution; Leica Biosystems, Nussloch, Germany), mouse anti-p53 monoclonal antibody (clone DO-7; 1:100 dilution; Agilent, Santa Clara, CA, USA), mouse anti-CD8 monoclonal antibody (clone; C8/144B; 1:100 dilution; Nichirei Biosciences, Tokyo, Japan), rabbit anti-CD4 monoclonal antibody (clone; SP35; 1:100 dilution; Cell-Marque, Rocklin, CA, USA), and mouse anti-FOXP3 monoclonal antibody (clone; 236AE/7; 1:100 dilution; Invitrogen). Samples were classified according to the intensity of Snail staining and scored as follows: 0, negative; 1, weak; 2, moderate; and 3, strong expression as shown in Fig. 1c. Samples with heterogeneous staining were scored based on intensity in the largest stained area. Scores of 0/1 and 2/3 were defined as Snail-low and Snail-high, respectively. Samples with heterogeneous staining were scored based on the intensity in the largest stained area. Tumor-infiltrating immune cells were evaluated as previously described in ref. [14]. Briefly, intratumoral CD33+ cells were counted from five fields at a magnification of ×200 and the average number of cells was calculated. CD8+, CD4+, and FOXP3+ cells were counted from five fields at a magnification of ×400 and the average number of cells was calculated.

**Flow cytometry**. For murine samples, mice with tumors were euthanized by $CO_2$ gas inhalation and spleens, femurs, tibias, and tumors were collected. Cells were stained with the following antibodies for 30 min at 4 °C. Antibodies used are listed in Supplementary Table 5. For IFNγ intracellular staining, a BD Cytofix/Cytoperm Fixation/Permeabilization Kit (BD Biosciences) was used. Non-viable cells were stained with 7-amino-actinomycin D (AAD) solution or DAPI solution and gated out. Matched isotype antibodies were used as controls. Data were acquired using a MACS Quant (Miltenyi Biotec) or FACS Calibur (BD Biosciences), and analyzed using MACS Quantify (Miltenyi Biotec) or Cell Quest Pro software (BD Biosciences)[14].

For human samples, ascites cells, ovarian tumor cells, and PBMCs were collected from patients with ovarian cancer at the time of surgery. Cells were stained with the following antibodies for 30 min at 4 °C. Antibodies used are listed in Supplementary Table 5.

The purified mouse MDSC (CD11b+Gr1+) population was sorted from subcutaneous tumor and ascites of mice inoculated with HM-1 cells using auto magnetic-activated cell sorting (autoMACS Pro separator; Miltenyi Biotech) according to the manufacturer's protocol. The purified human MDSC (CD11b+CD33+) population was sorted from patient ascites and ovarian tumor samples using CD33 MicroBeads (Miltenyi Biotech) according to the manufacturer's protocol.

**Chemotaxis assays**. In vitro migration of murine MDSCs was evaluated in 24-well plates with transwell polycarbonate-permeable supports (8.0 μm; Costar Corning,

Cambridge, MA, USA). MDSCs ($5 \times 10^5$; >80% purity) were plated in the upper chambers of the inserts 30 min after incubation with a CXCR2 antagonist at each concentration, and recombinant murine CXCL1, CXCL2, and CXCL5 (PeproTech) were placed in the lower chamber at a concentration of 1, 10, or 100 ng/mL. After incubation for 3 h, the number of MDSCs in the bottom compartment was counted using Acubright counting beads (Life Technologies, Carlsbad, CA, USA)[17].

To evaluate the migration of human MDSCs, MDSCs ($1 \times 10^5$; >80% purity) were plated in the upper chambers of the inserts 30 min after incubation with the CXCR2 antagonist at each concentration, and recombinant human CXCL1, CXCL2, and CXCL5 (PeproTech) were placed in the lower chamber at a concentration of 1 or 10 ng/ml. After incubation for 6 h, the number of MDSCs in the bottom compartment was counted[36].

**MDSC suppression assays**. An aliquot of the Gr1+ sorted cells was stained with an anti-CD11b antibody and analyzed by flow cytometry to ensure purity (>80%) of CD11b+Gr1+ cells. Carboxyfluorescein succinimidyl ester (CFSE; 10 μM) was added to suspensions ($1 \times 10^7$ cells/mL) of T cells separated from splenocytes of a wild-type B6C3F1 mouse using a mouse Pan-T-cell isolation kit (Miltenyi Biotech). Sorted MDSCs were added to CFSE-labeled T cells at different ratios, harvested in 96-well plates, and activated with Dynabeads Mouse T-activator CD3/28 (VER-ITAS, Tokyo, Japan) for 72 h at 37 °C. T-cell proliferation was evaluated by flow cytometry[14].

For human samples, CFSE was added to suspensions ($1 \times 10^7$ cells/mL) of T cells separated from PBMCs of healthy donors using a human Pan-T-cell isolation kit (Miltenyi Biotech). Sorted MDSCs from human ovarian tumor samples were added to CFSE-labeled T cells at different ratios, harvested in 96-well plates, and activated with Dynabeads Human T-activator CD3/28 (VERITAS, Tokyo, Japan) for 96 h at 37 °C.

**Clinical specimens of HGSOC patients**. Surgical specimens from patients with HGSOC who underwent primary surgery at Kyoto University Hospital between 1996 and 2014 were collected. We selected 56 advanced cases (48 stage III cases and 8 stage IV cases) for which samples of the peritoneal dissemination could be evaluated (Figs. 1, 3). The relevant clinical data were collected by retrospective review of patient files. Patients receiving chemotherapy or radiation prior to surgery were excluded.

**Clinical samples for CXCL1 and CXCL2 measurements**. Blood serum, surgical specimens, and PBMCs from 26 patients (9 stage I cases, 2 stage II cases, 9 stage III cases, and 6 stage IV cases) with primary ovarian cancer, who were treated at Kyoto University Hospital between 2012 and 2013, and blood serum from 8 healthy female donors were collected (Fig. 8). Patients receiving any prior treatment were excluded. The relevant clinical data were collected by retrospective review of patient files. To investigate Snail, CD33, CD8, CD4, and FOXP3 expression, we selected 12 advanced cases (9 stage III cases and 3 stage IV cases) for which samples of the peritoneal dissemination could be evaluated. Three out of all 6 stage IV cases had not undergone primary surgery. PBMCs from 9 advanced cases (6 stage III and 3 stage IV cases) were available (Supplementary Fig. 15). ELISA, immunohistochemistry analysis, scoring, and flow cytometry analysis were performed as described previously. CXCL1 and CXCL2 were evaluated by receiver operating characteristic curve analysis to define optimal cut-off levels for survival data (Fig. 8e, f).

**Clinical human patients' ascites**. Ascites cells from 13 patients with advanced ovarian cancer (8 stage III cases and 5 stage IV cases), hospitalized for massive cancerous ascites at Kyoto University Hospital between 2014 and 2015, were collected (Supplementary Fig. 14). Flow cytometry analysis was performed as described above.

**Bioinformatics analyses**. The TCGA HG-U133A dataset from the TCGA Data Portal (http://cancergenome.nih.gov) was analyzed by single-sample Gene Set Enrichment Analysis (ssGSEA) at the gene level, as previously described (Gene-Pattern version 3.5.1; http://www.broadinstitute.org/cancer/software/genepattern). Generic EMT scoring was used as previously described in ref. [3]. For data analyses, the ssGSEA scores were normalized from 0 to 1.

**Microarray analysis**. HM-1-control ($n = 3$), as well as HM-1-shSnail sh1 ($n = 2$) and sh2 ($n = 2$), cells were harvested. Whole RNA was extracted using the RNeasy Mini Kit (Qiagen), and was hybridized to an Affymetrix GeneChip® Mouse Transcriptome Array 1.0 (Affymetrix) and analyzed using the Affymetrix® Expression Console™ and Affymetrix® Transcriptome Analysis Console by GeneticLab (Hokkaido, Japan).

**Statistics**. All results were confirmed using at least three independent in vitro experiments or at least two independent in vivo experiments, and data from one representative experiment are presented. Results are shown as the average ± standard error of the mean (SEM). Comparisons between two groups were performed using Student's t-tests. Group comparisons were performed using one-way

ANOVAs. The log-rank test was used for overall survival analysis, and multivariate analysis was performed using Cox proportional hazards regression analysis. All statistical analyses were performed using GraphPad Prism 6 and Rcmdr software (http://www.rcommander.com/). The level of significance was set as *$P < 0.05$, **$P < 0.01$, ***$P < 0.001$, and ****$P < 0.0001$.

**Study approval**. All animal studies were approved by Kyoto University Animal Research Committee. Human studies were approved by Kyoto University Graduate School and Faculty of Medicine Ethics Committee. All human samples were acquired after obtaining written informed consent from each patient. The patients were identified by numbers.

**Data availability**. Microarray data have been deposited at Gene Expression Omnibus with accession number GSE109688. All other data are freely available from the corresponding author upon reasonable request.

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

## Acknowledgements

We thank Hana Ishiyama for her excellent technical assistance and J.B. Brown for his brilliant supervision regarding statistics. We also thank Prof. Harada and Dr. Katagiri for providing their pGL3/CMVmp plasmid. This work was supported by a Grant-in Aid for Scientific Research (KAKENHI) from the Ministry of Education, Culture, Sports, Science, and Technology (MEXT; No. 26253080 and No. 17K11275).

## Author contributions

M.T. designed and performed the experiments and wrote the manuscript; K.A. designed the experiments, provided funding, and edited the manuscript; T.B., J.H., and K.Y. designed the experiments; R.M. performed experiments and edited the manuscript; K.Y., N.H., and Y.H. performed the experiments; E.N. and A.S. designed and performed the experiments; M.M. and I.K. provided clinical samples and N.M. designed experiments, provided funding, and edited the manuscript.

## Additional information

**Competing interests:** E.N. was involved in this study as effort outside of the endowed associate professor of DSK project, Medical Innovation Center, Kyoto University sponsored by Sumitomo Dainippon Pharma Co Ltd. The remaining authors declare no competing interests.

