## [Peer Review File · Nature Communications]

Reviewers' comments:

Reviewer #1 (Remarks to the Author):

The manuscript by Mana Taki and co-authors investigates the link between Snail and CXCR2 ligands that favor the MDSC accumulation within tumor. The authors demonstrated that Snail-expressing tumor cells are able to activate the CXCL1 and CXCL2 transcription, optimizing the release of chemokines that support MDSC trafficking inside tumor. Molecularly, both NF- κ B-dependent pathway and direct Snail chromatin binding were required for chemokine release. As clinical connection, elevated levels of serum CXCL1 and CXCL2 not only were correlated with MDSC accumulation but also associated with poor prognosis in ovarian cancer patients.

While this work is novel and can have relevance for the understanding of immune mechanisms regulating cancer progression, there are several major issues that require attention:

1) The authors claim that Snail is correlated with unfavorable prognosis in ovarian cancer (Fig 1D). The authors should provide a multivariate analysis to dissect Snail as a negative, independent prognostic factor. Moreover, since Snail is linked with MDSC (at least CD33 positive cell) accumulation, also the tumor-infiltrating MDSCs amount could be a putative biomarker correlating with unfavorable prognosis. The author should add also this information in Figure 3. Finally, is there any evidence that also the amount of circulating MDSCs correlates with the Snail expression levels in tumor?

2) The authors claim that Snail induces EMT and promotes tumor progression in immunocompetent mice (Figure 2). The authors generated different mouse and human tumor models in which Snail expression is either abrogated or enforced, demonstrating that there was no alteration of in vitro cell proliferation that could be linked to Snail ablation. However, Snail also plays an essential role in conferring resistance to cell death; therefore, the authors should add data about indexes of cell death of tumor cells after the genetic engineering. About the in vivo models, it is difficult to understand why the authors selected two different cell administration routes in Fig. 2 and how they relate each other. Furthermore, the authors do not explain how they monitored the tumor growth in the intraperitoneally-inoculated mice and they did not indicate whether the reported data are representative of independent experiments or originate from a single experiment. Since the authors showed that PMN-MDSC expressed high level of CXCR2 and (likely) the tumor progression is linked to recruited MDSC through Snail-dependent control of chemokine release, the PMN-MDSC depletion using Ly6G-antibody should abrogate the tumor growth in immune competent mice. In this way the authors can reinforce the correlation of Snail-expression on tumor cells and the immune dysfunctions induced by MDSC accumulation. Finally, the authors should better characterize the adaptive immune response involved in tumor control: is it mediated by T cells or also NK cells have an impact on tumor control? The use of the immunodeficient mice does not dissect this point.

3) The authors claim that Snail is associated with increased intra-tumoral MDSCs (Fig.3). The authors did not provide data about immunosuppressive ability of tumor-infiltrating MDSCs (in mouse setting of solid tumor by subcutaneous inoculation), only the inverse correlation between tumor-infiltrating lymphocytes and MDSCs. Regarding human setting, it would be important to analyze the immunosuppressive activity of isolated CD33+ cells in different patients. In the case this is not possible, data about arginase-1 expression in Lin-CD33+HLADR- cells should be compared to Lin-CD33+HLADR+ and validated by enzymatic assay.

4) The authors claim that Snail induces CXCL1 and CXCL2 expression by the NF- κ B pathway (Figure 5). The author should describe better which NF- κ B pathway is activated by Snail (either canonical or non-canonical). Moreover, the authors should demonstrate the nuclear translocation of Snail.

5) The authors claim that CXCR2 ligands attract MDSCs through CXCR2 and promote tumor growth (Figure 6). The authors should better characterize/define what kind of human MDSCs express CXCR2: monocytic MDSC, PMN-MDSCs or immature MDSCs.

6) The authors claim that elevated levels of serum CXCL1 and CXCL2 reflect intra-tumoral MDSCs (Figure 8). Authors should provide information about patients and healthy donors (sex, Gender etc). Were there other leukocyte populations correlating with either CXCL1 or CXCL2 sera levels? It is also unclear how the cutoffs for CXCL1 and CXCL2 amounts were calculated.

Reviewer #2 (Remarks to the Author):

This manuscript submitted by Taki et al. described that Snail induces migration of tumor-infiltrating myeloid-derived suppressor cells through CXCR2 ligand upregulation in ovarian cancer. The results are highly interesting and the way data present is very clear. There are several comments that the authors can consider to improve the scientific quality of the current study.

1. Immunostaining in Fig. 1C raises an issue about the specificity of the antibody. It appears that the immunoreactivity is everywhere in score 2 and score 3. Should the positivity be observed in both tumor and stromal cells? The tumor cells with epithelioid features (unlikely undergoing EMT) are also positive. The Western blot of the antibody in ovarian cancer cells should be presented to show that the antibody only recognizes the Snail as a single band.

2. In Fig. 2A, the gel exposure time appears different in different panels.

3. In Fig. 3A, the case number (n= 3) is too low to come to a conclusion.

4. In Fig. 8E, the authors are in a good position to add more patients' samples for clinical outcome correlation.

5. In human tissue studies, the double staining of Snail and p53 should be done to indicate the EMT tumor cells in the stroma.

6. In this study, Snail-depleted (OVCAR8-shSnail) and Snail-overexpressing (OVCA433-Snail) human ovarian cancer cell lines were used. It would be better to use at least two cell lines for the KD approach. Likewise, an additional cell line should be applied for the over expression system.

RESPONSE TO REVIEWER #1:

We greatly appreciate your constructive comments concerning our manuscript entitled “Snail induces migration of tumor-infiltrating myeloid-derived suppressor cells through CXCR2 ligand upregulation in ovarian cancer”. We have studied your comments carefully and made major corrections which we hope will meet your approval. We have responded to your questions or comments in detail in the following text.

Detailed responses to the review:

Comment 1: *The authors claim that Snail is correlated with unfavorable prognosis in ovarian cancer (Fig 1D). The authors should provide a multivariate analysis to dissect Snail as a negative, independent prognostic factor. Moreover, since Snail is linked with MDSC (at least CD33 positive cell) accumulation, also the tumor-infiltrating MDSCs amount could be a putative biomarker correlating with unfavorable prognosis. The author should add also this information in Figure 3. Finally, is there any evidence that also the amount of circulating MDSCs correlates with the Snail expression levels in tumor?*

Response: We thank the reviewer for this pertinent comment.

First, in accordance with the reviewer’s comment, we provided a multivariate analysis as follows. We added the data in Supplemental Table 4 in the revised version.

	Univariate			Multivariate		
	RR	95% CI	P value	RR	95% CI	P value
Age \geq 55	2.22	1.03-4.45	0.045*	1.86	0.79-4.39	0.15
FIGO stage	1.86	0.73-6.70	0.17	0.37	0.076-1.77	0.21
Distant metastasis	2.62	1.27-13.16	0.020*	4.84	1.01-23.18	0.048*
Residual tumor	2.85	1.60-8.68	0.0027*	1.95	0.78-4.86	0.15
Snail expression	2.58	1.16-5.00	0.021*	2.79	1.10-7.05	0.031*

*statistical significance

We added the following text in *Results* (p. 6, lines 1–5) accordingly.

Multivariate analysis using Cox proportional hazards regression indicated that Snail expression in the peritoneal dissemination serve as a negative, independent prognostic factor for overall survival (relative risk, 2.79; 95% confidence interval (CI), 1.10-7.05; P = 0.031; Supplemental Table 4).

Second, we have previously reported that tumor-infiltrating MDSC count correlated with unfavorable prognosis in ovarian cancer (Horikawa N et al; *Clin Cancer Res*; 23; 587-599; 2017, Figure shown below). Therefore, we did not show the data in this manuscript. We have cited the information as follows in *Introduction* (p. 4, lines 1-2).

We previously reported that MDSC infiltration was inversely correlated with CD8+TIL numbers and shorter overall survival in advanced ovarian cancer.

Finally, we analyzed PBMCs of ovarian cancer patients, as shown in Figure 8 in the revised version. Nine PBMC samples were available. We provided Supplemental Figure 15, which shows that the amount of MDSCs in PBMCs was correlated with Snail expression in tumors and CXCL1 serum levels.

We therefore added the following text in *Results* (p. 16, lines 7-9).

MDSCs in PBMCs were also correlated with Snail expression in peritoneal dissemination samples and serum CXCL1 levels (Supplemental Figure 15).

Comment 2: The authors claim that Snail induces EMT and promotes tumor progression in immunocompetent mice (Figure 2). The authors generated different mouse and human tumor models in which Snail expression is either abrogated or enforced, demonstrating that there was no alteration of in vitro cell proliferation that could be linked to Snail ablation. However, Snail also plays an essential role in conferring resistance to cell death; therefore, the authors should add data about indexes of cell death of tumor cells after the genetic engineering. About the in vivo models, it is difficult to understand why the authors selected two different cell administration routes in Fig. 2 and how they relate each other. Furthermore, the authors do not explain how they monitored the tumor growth in the intraperitoneally-inoculated mice and they did not indicate whether the reported data are representative of independent experiments or originate from a single experiment. Since the authors showed that PMN-MDSC expressed high level of CXCR2 and (likely) the tumor progression is linked to recruited MDSC through Snail-dependent control of chemokine release, the PMN-MDSC depletion using Ly6G-antibody should abrogate the tumor growth in immune competent mice. In this way the authors can reinforce the correlation of Snail-expression on tumor cells and the immune dysfunctions induced by MDSC accumulation. Finally, the authors should better characterize the adaptive immune response involved in tumor control: is it mediated by T cells or also NK cells have an impact on tumor control? The use of the immunodeficient mice does not dissect this point.

Response: We deeply appreciate the reviewer's insightful comment.

First, in accordance with the reviewer's comment, we examined the indices of cell death of tumor cells by apoptosis assay using annexin-V and PI. Both deletion and overexpression of Snail did not affect cell apoptosis. We provided these data in Supplemental Figure 2E and added the following text in *Results* (p. 7, lines 6-7).

Altered Snail expression did not affect cell apoptosis (Supplemental Figure 2E).

Second, in this study, we used intraperitoneally inoculated mice as orthotopic ovarian cancer model. Although we regarded the survival reflected the tumor growth, we were unable to monitor tumor growth in early phase. Therefore, we mainly used subcutaneous-inoculated mice as in vivo model. To improve clarity of our manuscript, all data of intraperitoneally inoculated tumor models have been moved to Supplemental Figures in the revised version.

Third, we had previously not specified that the reported data are representative of at least 3 independent in vitro experiments or at least 2 independent in vivo experiments; we apologize for the oversight. The survival data of intraperitoneally inoculated mice is representative of two independent experiments with similar results. We have added detailed information in *Materials and Methods* (p. 34, lines 12-14).

Statistics

All results were confirmed using at least three independent in vitro experiments or at least two independent in vivo experiments, and data from one representative experiment are presented.

Fourth, in accordance with the reviewer's comment, we performed anti-Ly6G antibody treatment in immunocompetent mice. G-MDSC depletion using Ly6G-antibody suppressed tumor growth and intratumoral MDSC accumulation. We provided the data in Figure 7. Again, we highly appreciate the reviewer for this important suggestion.

We added the following text in *Results* (p. 14, lines 10–14).

In vivo G-MDSC depletion using an anti-Ly6G-antibody inhibited tumor growth in HM-1 tumor-bearing mice, but not in HM-1-shSnail tumor-bearing mice (Figure 7A). MDSCs significantly decreased and the CD8+T cell: MDSC ratio increased in anti-Ly6G antibody-treated HM-1-control tumors (Figure 7B). G-MDSC depletion did not affect intratumoral MDSCs of HM-1-shSnail tumor-bearing mice (Figure 7C).

Finally, the reviewer commented that adaptive immune response should be better characterized.

We counted the cell number of CD49b⁺ cells (mouse NK marker) in immunocompetent mouse tumors, which was higher in Snail-depleted tumors (Figure shown below). However, tumor growth in nude mice (with intact NK cells) was not affected by Snail depletion, indicating that the tumor growth inhibition is mainly mediated by CD8⁺ cells. We provided the data regarding the cell number of CD49b⁺ cells in immunocompetent mice tumor in Supplemental Figure 5.

We added the following text in *Results* (p. 8, lines 10-13).

CD49b⁺ cells (CD49b, a mouse natural killer (NK) cell marker) were increased in Snail-depleted tumors (Supplemental Figure 5), although Snail interference did not suppress tumor growth of nude mice with functioning NK cells (Figure 2E).

Comment 3: The authors claim that *Snail* is associated with increased intra-tumoral MDSCs (Fig.3). The authors did not provide data about immunosuppressive ability of tumor-infiltrating MDSCs (in mouse setting of solid tumor by subcutaneous inoculation), only the inverse correlation between tumor-infiltrating lymphocytes and MDSCs. Regarding human setting, it would be important to analyze the immunosuppressive activity of isolated CD33⁺ cells in different patients. In the case this is not possible, data about arginase-1 expression in Lin-CD33⁺HLADR⁻ cells should be compared to Lin-CD33⁺HLADR⁺ and validated by enzymatic assay.

Response: We analyzed immunosuppressive activity of tumor-infiltrating MDSCs sorted from subcutaneous mice tumors at least three times. Similar results were obtained in all experiments. We provided the representative data in Supplemental Figure 12A.

Furthermore, we also provided data showing the immunosuppressive ability of human CD33⁺ cells from ovarian tumors in two more patients in Supplemental Figure 7.

Comment 4: *The authors claim that Snail induces CXCL1 and CXCL2 expression by the NF- κ B pathway (Figure 5). The author should describe better which NF- κ B pathway is activated by Snail (either canonical or non-canonical). Moreover, the authors should demonstrate the nuclear translocation of Snail.*

Response: To describe which NF- κ B pathway is activated by Snail, we investigated RelB in nuclear protein, which is involved in the non-canonical pathway. RelB protein level was not changed by Snail depletion or Snail overexpression (data added in Figure 5A). Figure 5A shows that phosphorylated p65, an important factor in canonical NF- κ B pathway, was activated by Snail. Snail expression in the nucleus is confirmed by western blot in Figure 5A.

We described the data in *Results* (p. 12, lines 6-9):

RelB, a factor involved in the non-canonical NF- κ B pathway, was unchanged in Snail-depleted cell lines. The results suggest that the canonical NF- κ B pathway is activated by Snail.

Comment 5: The authors claim that CXCR2 ligands attract MDSCs through CXCR2 and promote tumor growth (Figure 6). The authors should better characterize/define what kind of human MDSCs express CXCR2: monocytic MDSC, PMN-MDSCs or immature MDSCs.

Response: We appreciate the reviewer's comment.

As shown in Supplemental Figure 8B, G-MDSCs mainly express CXCR2. Data for all patients were added to Supplemental Figure 8A.

We described the results as follows, in *Results* (p. 10, lines 1-3).

As expected, CXCR2 was highly expressed on MDSCs, especially G-MDSCs, in ascitic fluid samples from human ovarian cancer (Supplemental Figure 8, A and B).

We added the following text in *Discussion* (p. 18, lines 16–17).

Since human G-MDSCs mainly express CXCR2 (Supplemental Figure 8), a CXCR2 antagonist could be an effective treatment for ovarian cancer patients.

Comment 6: *The authors claim that elevated levels of serum CXCL1 and CXCL2 reflect intratumoral MDSCs (Figure 8). Authors should provide information about patients and healthy donors (sex, Gender etc.). Were there other leukocyte populations correlating with either CXCL1 or CXCL2 sera levels? It is also unclear how the cutoffs for CXCL1 and CXCL2 amounts were calculated.*

Response: We thank the reviewer for this constructive comment.

First, in accordance with the reviewer's comment, we provided more detailed information about patients and healthy donors in *Methods* (p. 32, line 17–p. 33, line 18).

Methods

Clinical samples for CXCL1 and CXCL2 measurements

Blood serum, surgical specimens, and PBMCs from 26 patients (9 stage I cases, 2 stage II cases, 9 stage III cases, and 6 stage IV cases) with primary ovarian cancer, who were treated at Kyoto University Hospital between 2012 and 2013, and blood serum from 8 healthy female donors were collected (Figure 8). Patients receiving any prior treatment were excluded. The relevant clinical data were collected by retrospective review of patient files. To investigate Snail, CD33, CD8, CD4 and FOXP3 expression, we selected 12 advanced cases (9 stage III cases and 3 stage IV cases) for which samples of the peritoneal dissemination could be evaluated. Three out of all 6 stage IV cases had not undergone primary surgery. PBMCs from 9 advanced cases (6 stage III cases and 3 stage IV cases) were available (Supplemental Figure 15). ELISA, immunohistochemistry analysis, scoring, and flow cytometry analysis were performed as described previously. CXCL1 and CXCL2 were evaluated by receiver operating characteristic (ROC) curve analysis to define optimal cut-off levels for survival data (Figure 8, E and F).

Clinical human patients' ascites

Ascites cells from 13 patients with advanced ovarian cancer (8 stage III cases and 5 stage IV cases), hospitalized for massive cancerous ascites at Kyoto University Hospital between 2014 and 2015, were collected (Supplemental Figure 14). Flow cytometry analysis was performed as described above.

Second, as the reviewer suggested, we examined the relationship between other leukocyte populations (CD8⁺, CD4⁺, and FOXP3⁺ cells) and CXCL1 or CXCL2 serum levels of the patients, as shown in Figure 8. We did not observe a correlation between other leukocytes and the levels of these chemokines. We provided the data in Supplemental Figure 14 and added the following text in *Results* (p. 16, lines 3-4).

There was no correlation between serum CXCR2 ligand levels and CD8⁺, CD4⁺ and FOXP3⁺ cells (Supplemental Figure 14).

A

B

Finally, we determined the optimal cut-off values based on the ROC curve analysis. The cut-off of CXCL1 was 42.83 pg/ml (AUC = 0.6842) whereas the cut-off of CXCL2 was 93.57 pg/ml (AUC = 0.6767). We provided the new survival data based on the optimal cut-offs and the information about the cut-offs in Figure 8.

RESPONSE TO REVIEWER #2:

We wish to express our strong appreciation to the Reviewer for the insightful comments on our manuscript entitled “Snail induces migration of tumor-infiltrating myeloid-derived suppressor cells through CXCR2 ligand upregulation in ovarian cancer”. They have helped us significantly to improve our manuscript. We studied the comments carefully and made major corrections which we hope will meet your approval. We responded to your questions or comments in details in the following text.

Detailed responses to the review:

Comment 1: *Immunostaining in Fig. 1C raises an issue about the specificity of the antibody. It appears that the immunoreactivity is everywhere in score 2 and score 3. Should the positivity be observed in both tumor and stromal cells? The tumor cells with epithelioid features (unlikely undergoing EMT) are also positive. The Western blot of the antibody in ovarian cancer cells should be presented to show that the antibody only recognizes the Snail as a single band.*

Response: We appreciate the reviewer’s comment.

First, we show the immunostaining of the positive control (human testis) and negative control (stained with control Rabbit IgG) as below. We confirmed the specificity of the antibody using both positive and negative controls.

Moreover, some reports have revealed that cells with epithelioid features have Snail expression (Bian XL, et al. Nat Commun. 2017 and Jaca A, et al. J Clin Pathol. 2017). Snail expression might be observed in stroma because tumor cells produce certain cytokines (e.g., TGF- β) to induce EMT in tumor-surrounding stroma.

Finally, we show the western blot data of anti-Snail antibody we used in ovarian cancer cells and HeLa (as a positive control).

We provided the data in Supplemental Figure 1 and referred to the specificity of the antibody in *Results* (p. 5, lines 17-18) as below.

We ascertained the specificity of the anti-Snail antibody we used (Supplemental Figure 1, A and B).

Comment 2: In Fig. 2A, the gel exposure time appears different in different panels.

Response: We thank the reviewer for this appropriate comment.

As the reviewer pointed out, the gel exposure time appears different in each panel. Therefore, we conducted western blotting again and presented the new data in Fig. 2A.

Comment 3: In Fig. 3A, the case number ($n=3$) is too low to come to a conclusion.

Response: In accordance with the reviewer's comment, we immunostained 6 cases each of Snail-depleted and control tumors. Because we decided to combine the in vivo models with the subcutaneous tumor models according to the recommendation from the other reviewer, we analyzed subcutaneous tumors and obtained similar results. We presented the data in Fig. 3A and moved all the data regarding intraperitoneally inoculated tumor models to Supplementary Figures in the revised version.

Comment 4: In Fig. 8E, the authors are in a good position to add more patients' samples for clinical outcome correlation.

Response: In accordance with the reviewer's comment, we added samples from 8 more patients to improve the reliability of results presented in Fig. 8A, B, E, and F. Data from 3 cases with peritoneal dissemination were added to Fig. 8C and D. We provided detailed information about patients in Fig. 8 in *Materials and Methods* (p. 32, line 12–p. 33, line 13), and revised *Results* (p. 16, lines 9-10) accordingly.

Results (p. 16, lines 6-7)

Snail expression showed a significant correlation with serum CXCL1 levels (Figure 8D).

Comment 5: *In human tissue studies, the double staining of Snail and p53 should be done to indicate the EMT tumor cells in the stroma.*

Response: In accordance with the reviewer's comment, we performed double staining of Snail (Red) and p53 (DAB) in human ovarian cancer samples, which is demonstrated below and in Supplemental Figure 1C to indicate EMT in ovarian cancer cells. p53⁺ tumor cells as well as some stromal cells expressed Snail in these cases. We added the following text in *Results* (p. 6, line 5).

p53⁺ ovarian tumor cells showed Snail expression (Supplemental Figure 1C).

Comment 6: *In this study, Snail-depleted (OVCAR8-shSnail) and Snail-overexpressing (OVCA433-Snail) human ovarian cancer cell lines were used. It would be better to use at least two cell lines for the KD approach. Likewise, an additional cell line should be applied for the over expression system.*

Response: We thank the reviewer for this comment. We included one more Snail-depleted (A1847-shSnail) and Snail-overexpressing (JHOS2-Snail) human ovarian cancer cell line each. RT-PCR analysis indicated that the expression of *CXCL1*, *CXCL2* and *CXCL5* was downregulated in Snail-depleted A1847-shSnail and upregulated in Snail-overexpressing JHOS2-Snail. These data are presented in Supplemental Figure 9.

We added the following text in Results (p. 10, lines 11-13) accordingly.

Two more human ovarian cancer cell lines (A1847-shSnail and JHOS2-Snail) were used to validate that the chemokine levels were affected by Snail, and we obtained similar results (Supplemental Figure 9).

Reviewers' comments:

Reviewer #1 (Remarks to the Author):

Editorial Note: Reviewer#1 expresses their satisfaction with the revision in their confidential comments to the editor.

Reviewer #2 (Remarks to the Author):

There is an issue concerning the Suppl Fig. 1C in the revised manuscript. The 'stromal' cells as shown have red immunoreactivity of SNAIL, but do not have p53 staining in nuclei. Moreover, these cells are smaller than the adjacent cohesive tumor cells. As a result, they may not represent tumor cells undergoing EMT. Do the authors have a better evidence to show this point which is critical for the entire study.

RESPONSE TO REVIEWER #2:

We would like to express our appreciation to the Reviewer for the valuable comment on our revised manuscript entitled “Snail induces migration of tumor-infiltrating myeloid-derived suppressor cells through CXCR2 ligand upregulation in ovarian cancer”. We studied the comment carefully and responded to your comment in details in the following text.

Comment: There is an issue concerning the Supplemental Fig. 1C in the revised manuscript. The 'stromal' cells as shown have red immunoreactivity of SNAIL, but do not have p53 staining in nuclei. Moreover, these cells are smaller than the adjacent cohesive tumor cells. As a result, they may not represent tumor cells undergoing EMT. Do the authors have a better evidence to show this point which is critical for the entire study.

Response: We thank the reviewer for the insightful comment.

In accordance with the reviewer’s comment, we observed again the ovarian cancer samples stained with p53 and Snail and provided the better images as follows.

There are small number of Snail⁺/p53⁺ tumor cells (→) at the invasive front. We would suggest that these cells might be undergoing EMT because they are losing apico-basal polarity.

As the reviewer pointed out, there are Snail⁺/p53⁻ stromal cells (▲) near the tumor-stroma interface. They are non-tumor stromal cells such as fibroblasts or immune cells. These cells have been previously described (Franci C et al.; *Oncogene*, 2006 and Blenchesmidt K et al.; *Br J Cancer*, 2008.); they would be comprised of cancer-associated fibroblasts (CAFs). They may represent Snail expression because these cells are often activated by tumor-producing cytokines such as TGFβ.

We are sorry to have confused the reviewer by what we have mentioned in our manuscript. In this manuscript, we have investigated a novel role of Snail as a mediator of immunosuppressive cells, but we were unable to detect whether EMT would have a direct impact on anti-tumor immunity.

Accordingly, we added the following text in Results and changed the data in Supplemental Figure

1C.

Results (p.5, line 18 – p.6, line 1)

We confirmed that p53⁺ ovarian cancer cells as well as small number of p53⁻ stromal cells expressed Snail (Supplemental Figure 1C).

Further, we added the following text as the one of our limitations in *Discussion*.

Discussion (p18, lines 4-5)

Although we did not find the direct impact of EMT to immune evasion,

REVIEWERS' COMMENTS:

Editorial Note:

Reviewer#2 comments to the editor that the new figure submitted is not convincing and suggest to delete this figure and its related discussion in the text.

RESPONSE TO REVIEWER #2 AND EDITOR

REVIEWERS' COMMENTS:

Editorial Note:

Reviewer#2 comments to the editor that the new figure submitted is not convincing and suggest to delete this figure and its related discussion in the text.

Thank you for the important suggestion. We deleted the Supplementary Fig. 1C and related sentences from the text;

Results (p5, line 18 – p6, line 1)

“We confirmed that p53+ ovarian cancer cells as well as small number of p53- stromal cells expressed Snail (Supplemental Figure 1C).

Discussion (p19, line 4 - 5)

“Although we did not find the direct impact of EMT to immune evasion”